# Decreased Blood Level of MFSD2a as a Potential Biomarker of Alzheimer’s Disease

**DOI:** 10.3390/ijms21010070

**Published:** 2019-12-20

**Authors:** María Sánchez-Campillo, María José Ruiz-Pastor, Antonio Gázquez, Juan Marín-Muñoz, Fuensanta Noguera-Perea, Antonio J. Ruiz-Alcaraz, Salvadora Manzanares-Sánchez, Carmen Antúnez, Elvira Larqué

**Affiliations:** 1Department of Physiology, School of Biology, Biomedical Research Institute of Murcia (IMIB-Arrixaca-UMU), University of Murcia, 30100 Murcia, Spain; medit2011@gmail.com (M.S.-C.); majoynerine@gmail.com (M.J.R.-P.);; 2Dementias´Unit, University Clinical Hospital “Virgen de la Arrixaca”, 30120 Murcia, Spain; juanmarinisla@gmail.com (J.M.-M.); fnogueraperea@gmail.com (F.N.-P.); salvi.manzanares@gmail.com (S.M.-S.); mcarmen.antunez@carm.es (C.A.); 3Department of Biochemistry and Molecular Biology B and Immunology, Faculty of Medicine, Biomedical Research Institute of Murcia (IMIB-Arrixaca-UMU), University of Murcia, 30120 Murcia, Spain; ajruiz@um.es

**Keywords:** aging, neurologic disorders, omega-3 PUFA, MFSD2a carrier, Alzheimer’s disease

## Abstract

The protein Major Facilitator Superfamily Domain containing 2A (MFSD2a) was recently described as the primary carrier for docosahexaenoic acid (DHA) into the brain. Alzheimer’s disease (AD) is a progressive neurodegenerative disorder characterized by lower DHA levels in blood lipids. The aim of this study was to investigate the expression of MFSD2a in the whole blood and brain as a potential biomarker of AD. Three groups were established: 38 healthy controls, 48 subjects with moderate AD (GDS4), and 47 with severe AD (GDS6). We analyzed postmortem brain samples from the hippocampus of 11 healthy controls and 11 severe AD patients. Fatty acid (FA) was determined in serum and brain by gas chromatography. Blood and brain MFSD2a protein expression was analyzed by Western blotting. We found a significant and progressive decline of MFSD2a levels in blood of AD patients (Control 0.83 ± 0.13, GDS4 0.72 ± 0.09, GDS6 0.48 ± 0.05*, *p* ˂ 0.01). We also corroborated a significant reduction of DHA and other n-3 long-chain polyunsaturated FA in serum of AD. No differences were found in MFSD2a expression or FA levels in brain of controls and AD subjects. MFSD2A carrier was analyzed in AD patients for the first time and the level of MFSD2a in the whole blood could be a potential biomarker of this disease.

## 1. Introduction

Alzheimer’s disease (AD) is a progressive, irreversible, neurodegenerative disorder and one of the most common causes of dementia in old age. AD currently affects about 35 million people in the world [1]. The Global Deterioration Scale (GDS) provides an indication of the seven different stages of cognitive function for AD patients. Stages 1–3 are the pre-dementia period and stages 4–7 correspond to the dementia step. At the beginning of stage 5, the individual needs assistance [2]. The neurodegeneration produced in AD might originate from the accumulation of amyloid β-peptide (Aβ) in the brain, and it is the primary influence driving AD pathogenesis. The rest of the disease process, including formation of neurofibrillary tangles containing tau protein, has been proposed as the result from an imbalance between Aβ production and clearance [3]. In AD, the ability of synapses to transmit information is reduced, the number of synapses is decreased, and then death of the neurons occurs [4]. High-fat Western diets seem to promote the progression of AD-like pathology through enhancement of cerebral myeloid angiopathy and oxidative stress [5], but the leading mechanisms are still unclear.

Mediterranean lifestyle patterns could be beneficial for AD since dietary intake of n-3 fatty acids (FA) and weekly consumption of fish may reduce the risk of AD [6,7]. The more abundant n-3 polyunsaturated fatty acids (PUFA) in fish oil, docosahexaenoic acid (DHA; 22:6n-3), and eicosapentaenoic acid (EPA; 20:5n-3), are proposed to be beneficial components in the prevention of AD [8]. DHA could exert protective effects against β-amyloid production, accumulation, and potential downstream toxicity [6]. DHA is implicated in a diversity of physiological processes, as well as memory formation, aging, synaptic membrane function, biogenesis, function of photoreceptors, and neuroprotection [9]. A diet enriched with DHA reduces amyloid burden in an aged Alzheimer animal mouse model [10]. Nevertheless, n-3 PUFA (polyunsaturated fatty acids) supplementation might not be useful in the advanced stages of AD [11] in which significant neuronal failure has already occurred [8]. The meta-analysis from de Wilde et al. reported significantly lower levels of DHA in blood and brain from AD patients [12]. Thus, mechanisms to improve DHA in brain of AD are of major interest.

Given that DHA synthesis in brain is limited [13], the uptake from serum PUFA plays a critical role in the maintenance of brain lipid composition in neurological diseases [14]. The blood–brain barrier (BBB) is a specialized multicellular membrane of astrocytes, pericytes, and endothelial cells that control the selective uptake of molecules into the brain and keep away toxins and pathogens. BBB provides an optimal environment for brain function [15]. Recently, a transmembrane protein named Major Facilitator Superfamily Domain containing 2A (MFSD2a) was described as the primary carrier for the DHA uptake, and other long-chain FA such as lyso-phospholipids (Lyso-PL) in BBB [16] and in retinal pigment epithelium [17]. MFSD2A is expressed by the endothelial cells and presents 12 transmembrane domains composed by two evolutionary duplicated six transmembrane units [18]. MFSD2a is a carrier with a dual role in the brain, the uptake of unsaturated lyso-PL as DHA, and the establishment of BBB integrity by the inhibition of caveolae-mediated transcytosis [16,19].

Humans with homozygous mutations that inactivate the MFSD2a gene present intellectual disability and severe microcephaly [20,21,22]. Furthermore, MFSD2a knock-out mice showed reduced levels of DHA in brain, deficits in learning and memory, neuronal cell loss in hippocampus and cerebellum, and severe microcephaly [16]. Additionally, in a mouse model it has been observed that MFSD2a is required at the BBB for normal postnatal brain growth and for the maintenance plasma membrane phospholipid composition [23].

On the other hand, short-term fish oil treatment in AD mice did not affect the expression of MFSD2a in the brain, while it tended to be enhanced in the liver [24]. Due to its important role in the central nervous system (CNS) and the BBB physiology, MFSD2a means an interesting issue to study in neurodegenerative disorders, such as AD. It is unknown whether MFSD2a is been altered somehow in AD or if its level in blood could be a potential indicator of the AD. Since it is so difficult to evaluate the expression of MFSD2a in the brain of AD patients in vivo, new biomarkers based on MFSD2a that could be analyzed in easily-to-obtain samples such as blood should be explored.

We aimed to study whether MFSD2a level is altered in the blood of AD patients, which are easy-to-obtain samples by noninvasive procedures, on different stages of the disease as a potential biomarker of AD. In addition, we studied in a small subset of samples the MFSD2a levels and fatty acid profiles in postmortem brain samples from AD and control subjects.

## 2. Results

### 2.1. Demographic Characteristics of Subjects

The characteristics of the 133 participants are described in Table 1. Although there were slight differences in age between groups, body mass index (BMI) did not differ significantly between AD patients and controls (Table 1). Over 50% of AD patients suffered from hypertension. GDS6 indicated cognitive severe impairment, while GDS4 corresponded to middle cognitive impairment. No differences were found between the two groups of AD patient in regard to diabetes and hypertension. However, there was a trend towards higher percentages of smoking in the GD6 group of Alzheimer’s patients. In the case of the Control group, these and other demographic information were not available due to a limitation of data recording in the biobank data provided from such samples. 

### 2.2. Reduced Level of Blood MFSD2a in Alzheimer’s Disease Patients and Impact on Fatty Acid Profile in Serum

We reported for the first time a continuous decline of MFSD2a protein level in blood of patients with different grades of AD, the differences being statistically significant between GDS6 patients and controls (Figure 1).

There was a significantly higher concentration of total FA in serum of both GDS4 and GDS6 AD groups with respect to controls (Table 2). Stearic acid (18:0) and other minor saturated FA (22:0 and 23:0) decreased in serum of patients with AD, but that was not the case for the totality of all saturated FA. However, the percentages of n-3 PUFA and LC-PUFA were significantly decreased in serum of AD in GDS6, while there were no differences in neither n-6 PUFA nor n-6 LC-PUFA percentages among the three experimental groups. Therefore, the ratio n-6/n-3 PUFA increased significantly in the GDS6 group of AD patients.

The decrease in n-3 LC-PUFA in serum was due to lower percentages of both EPA and DHA. There was a not significant but clear trend towards a decrease of the DHA percentage in serum of GDS4 and GDS6 groups (*p* = 0.062) (Figure 2a); in fact, the statistical *t*-test analysis showed a significant reduction of the DHA percentage in serum controls vs. grouped AD subjects (both GDS4 + GDS6) (3.54 ± 0.18 vs. 3.06 ± 0.11*, *p* = 0.02). Concerning EPA percentage, we found lower percentages in AD than in controls (Figure 2b). On the other hand, MFSD2a level in blood did not correlate either with serum DHA (r = –0.68, *p* = 0.453) or EPA percentage (r = 0.017, *p* = 0.853). 

### 2.3. Levels of MFSD2a Expression in Brain Stay Unaltered in Alzheimer’s Disease Patients

We also analyzed MFSD2a level in a small set of brain samples from other postmortem subjects (Control *n* = 11, GDS6 *n* = 11) (Figure 3). No differences were found in MFSD2a levels in the brain, and even MFSD2a tended to increase in the AD group (Figure 3). FA profile in brain did not change with AD. DHA, EPA, and the n-6/n-3 ratio were lower in AD GDS6, but did not reach statistical significance, probably due to the low number of analyzed samples (Table 3). MFSD2a level in the brain did not correlate with brain DHA percentage (r = –0.74, *p* = 0.745).

## 3. Discussion

This is the first study that has analyzed MFSD2a levels in the blood of humans under different stages of AD. The participants of this study were non-supplemented with n-3 PUFA. We demonstrated for the first time that MFSD2a protein expression decreased significantly in blood samples obtained from AD patients compared to healthy subjects. Nevertheless, the reduction of MFSD2a protein expression observed in blood was not accompanied by the correspondent reduction in the brain, which could indicate that MFSD2a protein level in the brain could be differently regulated in AD. Low levels of MFSD2a in the whole blood could affect peripheral tissue functions in AD patients, which should be explored in future studies.

AD has been associated with metabolic disorders such as obesity, hypertension, hypercholesterolemia, and diabetes [26,27]. In the present study, we found a significantly higher concentration of total lipids in AD (Table 2) and more than 50% of AD patients suffered from hypertension, while nearly 26% were diabetics (Table 1), which could explain the higher levels of total lipids in serum of AD patients, although this point should be further explored by comparison with more complete records of data from control subjects.

A significant decline in the percentage of n-3 LC-PUFA was observed in serum concomitantly to the progression and severity of AD (Table 2). This result is related with the decrease of both DHA and EPA percentages in the serum of AD groups (Figure 2). In agreement with previous studies [24,28], the decrease of n-3 FA percentages in serum leads to a significant increase in the n-6/n-3 PUFA ratio (Table 2). These data are consistent with previous studies performed in old people with dementia that reported also lower concentrations of plasma n-3 PUFA [12,26,29]. A disturbed n-6/n-3 ratio is related with inflammation and some neurological diseases, and several studies have described an improvement of n-6/n-3 ratio after short-term fish oil supplementation [24,30]. Many factors such as diet, nutrient absorption, metabolic disturbances, or the increased use of nutrients during the processes related to the AD pathology may alter nutrient levels in the plasma of these patients [26]. Furthermore, subjects receiving DHA supplements were excluded in the present study. Although we did not evaluate dietary intake of patients, the clear decline of DHA levels already in GDS4 subjects point towards AD as the most probable cause of the lower serum n-3 FA levels in such patients, due to an impaired systemic availability of numerous nutrients [26]. In addition, Wang et al., proposed that high oxidative stress in AD patients may decrease PUFA level, since they can suffer lipid peroxidation and break down into toxic compounds, like malonaldehyde. In fact, they postulated DHA as a potential biomarker of AD, because its alterations could reflect metabolic changes taking place during AD development [31].

In the present study, we demonstrated a significant continuous decline of MFSD2a protein level in blood of patients with AD (Figure 1). This suggests that blood cells, easily obtainable through peripheral blood sampling, express the MFSD2a carrier and that its expression could be related with AD. This result would be in line with the behavior of other FA carriers in blood of AD patients such as fatty acid translocase (FAT/CD36), whose expression levels also decreased in peripheral leukocytes of these patients [32]. Apart from being a FA carrier, FAT/CD36 has an important role as a scavenger receptor that recognizes amyloid plaques, and has been reported to trigger oxidant production by macrophages and microglia [33]. Our results suggest that altered MFSD2a levels in blood might inform us about metabolic disorders and/or the nutrient transport across other human tissues, which ought to be studied in future.

In this research, both MFSD2a protein level in blood and DHA percentage in serum decreased in advanced stages of AD, although the blood level of MFSD2a did not lineally correlate with the serum level of DHA observed. Despite the differences in age between Control and AD groups, patients from GDS4 and GDS6 had similar age but a gradient to lower MFSD2a and DHA levels in blood respect to controls. Thus, the pathology seems to have the highest effect on the parameters analyzed. MFSD2A plays different roles, as it is a carrier of lipid metabolism, but also participates in body growth and development, and motor function; and its codifying gene is nutritionally regulated in mice [34]. MFSD2a is not an exclusive carrier of DHA, because can also transport other ligands [21]; so, the decrease of MFSD2a levels in the advanced stages of AD might not be necessarily parallel the decline of DHA levels. While, other mechanisms apart from blood MFSD2a may be contributing as well to the decreased percentage of DHA observed in the serum of AD patients. In addition, it is important to highlight that the decline of MFSD2a levels in blood is more acute and occurs at earlier stages of AD than the reduction of DHA levels in serum. Therefore, MFSD2a could be postulated as a more sensitive biomarker to diagnose different stages of AD in easy-to-obtain blood samples, without the necessity of applying other more invasive methods. 

We also studied MFSD2a expression in the hippocampus in another small set of postmortem subjects with GDS6, but no statistical differences were found compared to controls (Figure 3). Thus, MFSD2a expression protein seems to be strongly regulated in the hippocampus of AD patients not supplemented with n-3 PUFA. Our results were similar to those obtained by Milanovic et al. [24], who reported no changes in MFSD2a in both liver and brain in an AD mouse model after a fish oil supplementation period. However, in the liver, strong trends of enhanced MFSD2a protein expression was observed by supplementation with fish oil for a short term (3 weeks), while no significant increases of MFSD2a were found in the brain of AD models [24]. Although our low number of brain samples might have limited the power of the analysis, it is also possible that the higher expression of MFSD2a observed in brain could try to act as a compensatory mechanism in an attempt to improve DHA accretion into this tissue. Berger et al., reported that the expression of MFSD2a in brain is ubiquitous and slowly induced [34]. Thus, it appears to be difficult to modify MFSD2a levels in the brain. On the other hand, Sandoval et al. demonstrated opposite differences of MFSD2a expression between the cortical and subcortical regions in healthy animals after long term supplementation with different FA, probably due to the different cellular composition and nutrient transport requirements of each brain region [35]. We analyzed MFSD2a level and FA profile only in the hippocampus, but not in the cortex or any other areas of the brain. Thus, the study of other brain regions could report additional information on the regulation of MFSD2a expression in the whole brain. Changes in brain components are very complex and would require many further studies. Regrettably, it was not possible to analyze MFSD2a levels in blood of such postmortem patients because the corresponding blood samples were not available in the biobank.

DHA has a significant role in the hippocampus, which is critical for learning [10]. E-series resolvins (RvEs) derived from eicosapentaenoic acid (EPA), and D-series resolvins (RvDs), protectin/neuroprotectin (PD/NPD) and maresins (MaRs) derived from DHA have local potent pro-resolving actions [36,37,38]. We did not find significant differences in the percentages of DHA or EPA in the hippocampus of the postmortem subjects between controls and GDS6 AD patients, probably due to the low number of samples available (Table 3). A meta-analysis reported significantly lower levels of DHA in the brain of AD patients [12], although six studies of such meta-analysis reported significantly lower levels of DHA in AD patients than in controls, while seven studies reported no significant differences between groups [12]. In the OmegAD study in AD patients, 6 months of supplementation with 2.3 g of DHA enhanced both EPA and DHA in cerebrospinal fluid and plasma [39]. However, changes of EPA and n-3 docosapentaenoic acid (22:5) in cerebrospinal fluid and plasma were strongly correlated. In contrast, DHA in cerebrospinal fluid and plasma were not correlated [39]. Despite the existence of a disturbed uptake of DHA respect to EPA in BBB in AD patients, a different metabolization of these two FA within the brain could also occur. Regardless, the changes in DHA levels in cerebrospinal fluid were inversely related to phosphorylated tau protein and inflammatory biomarkers, which highlight the importance to have appropriate DHA levels in the brain of someone with AD [39].

We could not find a correlation of MFSD2a level with DHA and other FA percentages in the hippocampus. This suggests that other mechanisms apart from MFSD2a are contributing to DHA percentage in the serum and the brain of AD patients. It was previously reported, in an AD animal model, that a short-term fish oil supplementation did not modify DHA levels in the brain of AD animals, while in wild type animals both EPA and DHA were increased. Instead, in the liver, fish oil increased DHA in both AD and wild type animals [24]. Thus, the n-3 fatty acid profile is strongly protected in the brain compared with other tissues, especially in AD patients. 

In conclusion, MFSD2a carrier is expressed in blood and its expression is reduced concomitantly with the progression of AD. Therefore, MFSD2a protein level in blood could be an additional potential biomarker of AD progression. The function of MFSD2a and localization in blood needs further clarification.

## 4. Materials and Methods 

### 4.1. Study Population

All samples were provided by the “Biobanco en Red de la Región de Murcia”, BIOBANC-MUR, integrated in the Spanish Biobanks Network Platform (www.redbiobancos.es)). The study protocol was approved by the Hospital Ethics Committee (Approval code 2016-10-6-HCUVA; 31 October 2016) in accordance with the Declaration of Helsinki. Three groups of subjects were established from age 52 to 84 years old: 38 healthy controls (GDS1), 48 subjects diagnosed of moderate AD (GDS4), and 47 subjects diagnosed of severe AD (GDS6). Sample size was estimated based on DHA percentages in plasma PLs of subjects published by Cunnane et al. [40]. Type I error was set at α = 0.05 and type II error β = 0.2 (power 80%), obtaining a minimum sample size of nine subjects per group. The software used for this estimation was nQuery 7.0 (Statsols HQ, Cork, Ireland).

This sample was described in relation to the Global Deterioration Scale (GDS) developed by Dr. Barry Reisberg, which provides to caregivers an overview of the stages of cognitive function for those suffering from a primary degenerative dementia such as Alzheimer’s disease. It is broken down into seven different stages. Stages 1–3 are the pre-dementia stages. Stages 4–7 are the dementia stages. Beginning in Stage 5, an individual can no longer survive without assistance [2].

### 4.2. Western Blotting Analyses

Protein extracts were obtained from 15 µL of whole blood by adding 285 µL of cell lysis buffer (Cell Signaling Technology, Danvers, MA, USA) containing 1 mM PMSF (Sigma Aldrich, Saint Louis, MO, USA). Samples were homogenized in a TissueLyser LT (QIAGEN, Barcelona, Spain) for 3 minutes and centrifuged at 13,000 rpm 4 °C for 15 min. Protein extracts from 100 mg of brain were obtained as previously detailed [41]. Protein was quantified by Bradford assay [42]. Samples were stored at –80 °C until Western blotting analysis.

### 4.3. Antibodies

Primary rabbit polyclonal antibodies against MFSD2a, (Abcam, Cambridge, UK. Ref.: ab105399) previously validated by several steps [23], β-actin (Sigma Aldrich, Saint Louis, MO, USA) and GADPH (Abcam, Cambridge, UK. Ref.: ab8245) were used. Polyclonal anti-rabbit secondary antibodies conjugated with horseradish peroxidase were obtained from Santa Cruz Biotechnology (Santa Cruz Biotechnology, Santa Cruz, CA, USA).

Protein extracts (30 µg protein from postmortem brain samples from hippocampus and 11 µg from blood) diluted in sample buffer and protein standard (Dual Color Precision Plus Protein Standars, Biorad, Richmond, CA, USA) were resolved on 10% polyacrylamide gels, and transferred into polyvinylidene difluoride membranes (Millipore Corporation, Danvers, MA, USA), which were then blocked in PBS-T (phosphate saline buffer with 0.1% Tween-20) containing 2% BSA for 1 h at room temperature. Thereafter, membranes were incubated with anti-MFSD2A antibodies 1:1000 overnight at 4 °C. Blots were washed with PBS-T and probed for 1 h at room temperature with the corresponding secondary antibody conjugated with horseradish peroxidase. Finally, membranes were stripped with Tris/HCL-Buffer pH 2.3 containing 0.1 M β-mercaptoethanol (Sigma Aldrich, Saint Louis, MO, USA) and retested with antibodies to GADPH 1:1000 for blood or β-actin 1:15,000 in brain samples as loading control. To quantify the level of MFSD2a in the samples, we run 12 samples in each gel, plus another sample as calibrator, with respect to their loading control [43,44]. Proteins were detected with a chemiluminescence kit following the manufacturer’s instructions (Pierce ECL 2 Western blotting Substrate; Thermo Scientific, IL, USA). Densitometry was performed on all blots to determine the density of the bands, using AmershamTM Imager 600RGB (GE Healthcare, Barcelona, Spain) and the software ImageQuant TL version 8.1 (GE Healthcare, Barcelona, Spain). MFSD2A relative protein level data were normalized against β-actin level in the case of brain or GADPH level for blood.

### 4.4. Analysis of Fatty Acids 

Total FA from serum (250 µL) and brain (30 mg) were extracted in chloroform:methanol (2:1, *v*/*v*) according to Folch et al. [45]. Before the extraction, 0.05 mg of pentadecanoic acid was added as internal standard. Synthesis of FA methyl esters was performed according to Stoffel et al. [46] by adding 1 mL of 3 N methanolic HCl (Sigma Aldrich, Saint Louis, MO, USA) to lipid extracts and heating at 90 °C for 1 h: derivatives were extracted into hexane and stored at –20 °C until gas chromatographic analysis. We used a SP-2560 capillary column (100 m × 0.25 mm × 20 µm) (Sigma Aldrich, Saint Louis MO, USA) in a Hewlett-Packard 6890 gas chromatograph (Agilent Technologies, Santa Clara, CA, USA). The temperature of the detector and the injector was kept at 240 °C. The oven temperature was programmed 30 min at 175 °C and was increased at a rate of 5 °C per min to 230 °C for 17 min. Helium was used as the carrier gas at a pressure of 45 psi. Peaks were identified by comparison of their retention times with appropriate standards (Sigma Aldrich, Saint Louis, MO, USA).

We provide the fatty acid profile as percentage values of each fatty acid respect to the total fatty acids. In addition, the absolute concentration of total fatty acids in the samples is specified at the end of the tables. Multiplying the percentage of the specific fatty acid by the total amount of fatty acids can be estimated the absolute concentration of each fatty acid analyzed.

### 4.5. Statistical Analysis

The results are expressed as mean ± standard error of the mean (SEM). Data followed a normal distribution according to Shapiro–Wilk test. Differences between Control, GDS4, and GDS6 groups were evaluated by ANOVA followed by post hoc Bonferroni analyses. Differences between control and all AD subjects were analyzed by Student’s *t*-test. Qualitative data as sex, hypertension, diabetes, smoker, and family background were analyzed by chi-square. The significance level was set at *p* < 0.05. Pearson correlation analyses were also performed. The statistical analyses were evaluated by the SPSS® 16.0 software package (SPSS, Chicago, IL, USA). 

## 5. Conclusions

This is the first study that has analyzed MFSD2a protein level in the human blood under different stages of Alzheimer’s Disease. MFSD2a carrier is expressed in blood and its expression is reduced concomitantly with the progression of AD. Therefore, MFSD2a protein level in blood could potentially be an additional early biomarker of AD progression. The function of MFSD2a and localization in blood need further clarification.

## Figures and Tables

**Figure 1 ijms-21-00070-f001:**
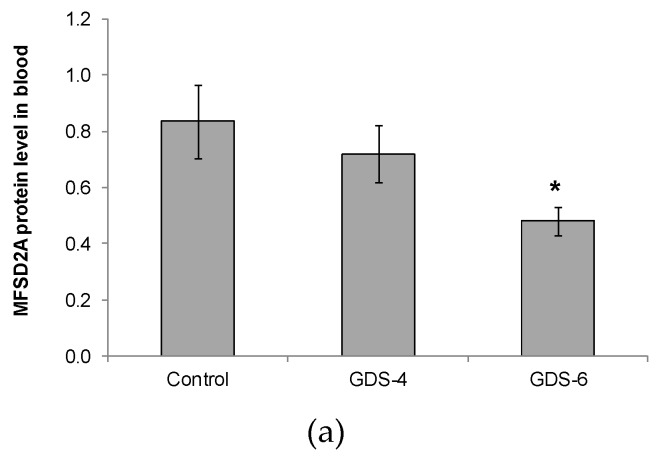
(**a**) Relative protein level of Major Facilitator Superfamily Domain containing 2A (MFSD2a) in the whole blood of Control, GDS4, and GDS6 groups (*p* = 0.039). Results are expressed as mean ± SEM. ANOVA followed by Bonferroni test was used to assess differences between the groups. Significant differences are indicated by footnote symbols: * Result indicates a significant difference compared to the Control group (*p* < 0.05); † Result would indicate significant differences compared to the GDS4 group (*p* < 0.05). (**b**) Example of Western blot analysis of MFSD2a and D-Glyceraldehyde-3-Phosphate Dehydrogenase (GADPH) expression in blood from controls and Alzheimer’s disease (AD) subjects.

**Figure 2 ijms-21-00070-f002:**
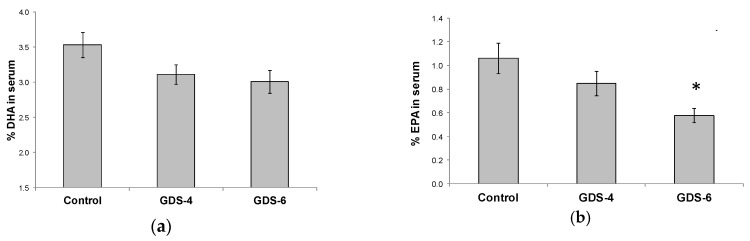
(**a**) Docosahexaenoic acid (DHA) percentage in serum of the Control, GDS4, and GDS6 groups (*p* = 0.062). (**b**) Eicosapentaenoic acid (EPA) percentage in serum of the Control, GDS4, and GDS6 groups (*p* = 0.004). Results are expressed as mean ± SEM. ANOVA followed by Bonferroni test was used to assess differences between the groups. Significant differences are indicated by footnote symbols: *Result indicates a significant difference compared to the Control group (*p* < 0.05); † Result would indicate significant differences compared to the GDS4 group (*p* < 0.05).

**Figure 3 ijms-21-00070-f003:**
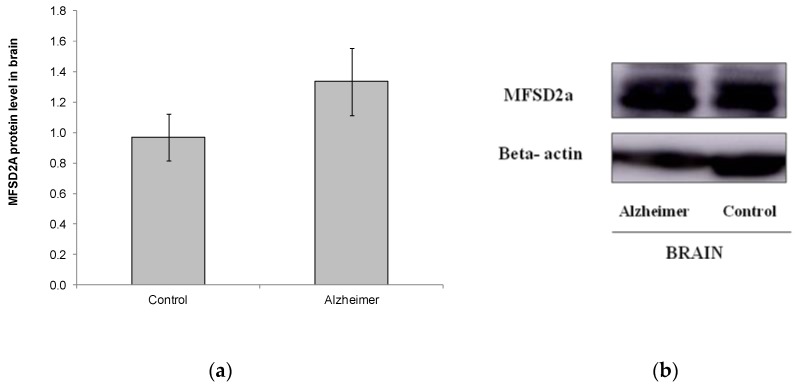
(**a**) Relative protein level of MFSD2a in the brain of controls and AD patients (*p* = 0.183). Results are expressed as mean ± SEM. A Student’s *t*-test was used to assess differences among the groups. (**b**) Example of Western blot analysis of MFSD2a and GADPH expression in postmortem brain tissue from control and GDS6 of AD patients. Significant differences are indicated by footnote symbols: * Result indicates a significant difference compared to the Control group (*p* < 0.05); † Result would indicate significant differences compared to the GDS4 group (*p* < 0.05).

**Table 1 ijms-21-00070-t001:** Demographic characteristics of subjects.

	Control *n* = 38	GDS4 *n* = 48	GDS6 *n* = 47	*p*
Sex (*n*)	10♂, 28♀	13♂, 35♀	18♂, 29♀	0.385
Age	67 ± 2	74 ± 1*	76 ± 1*	**<0.001**
BMI (kg/m^2^)	27.4 ± 0.8	26.2 ± 1.8	28.9 ± 1.4	0.300
Hypertension (%)	n/a	50.0	51.1	0.917
Diabetes (%)	n/a	27.1	26.1	0.913
Smoking (%)	n/a	14.6	25.5	0.182
Dementia family history (%)	n/a	73.9	68.3	0.563
MMSE score	n/a	18.7 ± 0.7	8.3 ± 0.8 **^†^**	**<0.001**

Results are expressed as mean ± SEM or as percentage. MMSE, Mini Mental State Exam [25]. Significant differences are indicated by footnote symbols: * Result indicates significant difference compared to the Control group (*p* < 0.05); † Result indicates significant differences compared to the GDS4 group (*p* < 0.05) (bold face); n/a: Not available

**Table 2 ijms-21-00070-t002:** Fatty acid profile (%) in serum of Alzheimer’s disease patients and control subjects.

Fatty Acid (%) with Respect to Total Fatty Acid Content	Control (*n* = 38)	GDS4 (*n* = 48)	GDS6 (*n* = 45)	*p*
Lauric acid (12:0)	0.12 ± 0.03	0.12 ± 0.02	0.14 ± 0.02	0.843
Myristic acid (14:0)	1.49 ± 0.19	1.51 ± 0.10	1.54 ± 0.15	0.971
Palmitic acid (16:0)	21.79 ± 0.30	21.74 ± 0.29	22.33 ± 0.28	0.282
Hexadecanoic acid (16:1 n-9)	0.29 ± 0.01	0.31 ± 0.01	0.31 ± 0.01	0.338
Palmitoleic acid (16:1 n-7)	1.37 ± 0.12	1.45 ± 0.09	1.63 ± 0.12	0.240
Margaric acid (17:0)	0.18 ± 0.02	0.22 ± 0.01	0.23 ± 0.01*	**0.040**
Stearic acid (18:0)	8.39 ± 0.21	7.62 ± 0.18*	7.72 ± 0.18*	**0.012**
Oleic acid (18:1 n-9)	20.19 ± 0.63	21.10 ± 0.50	20.62 ± 0.56	0.525
Cis-vaccenic acid (18:1 n-7)	1.59 ± 0.05	1.67 ± 0.04	1.75 ± 0.06	0.073
Linoleic acid (18:2 n-6)	26.64 ± 0.76	26.94 ± 0.76	26.55 ± 0.77	0.931
γ-linolenic acid (18:3 n-6)	0.40 ± 0.03	0.42 ± 0.02	0.43 ± 0.03	0.755
α-linolenic acid (18:3 n-3)	0.16 ± 0.02	0.19 ± 0.01	0.16 ± 0.01	0.319
Arachidic acid (20:0)	0.30 ± 0.02	0.31 ± 0.03	0.31 ± 0.02	0.971
Gondoic acid (20:1 n-9)	0.23 ± 0.03	0.29 ± 0.03	0.27 ± 0.03	0.392
Eicosadienoic acid (20:2 n-6)	0.21 ± 0.03	0.40 ± 0.03*	0.32 ± 0.04*	**<0.001**
Dihomo-γ-linolenic acid (20:3 n-6)	1.63 ± 0.06	1.65 ± 0.05	1.59 ± 0.06	0.738
Arachidonic acid (20:4 n-6, AA)	7.29 ± 0.29	7.20 ± 0.28	7.71 ± 0.33	0.445
Heneicosylic acid (21:0)	0.05 ± 0.01	0.06 ± 0.01	0.03 ± 0.01	0.073
Behenic acid (22:0)	0.73 ± 0.04	0.65 ± 0.03	0.63 ± 0.02*	**0.046**
Adrenic acid (22:4 n-6)	0.08 ± 0.02	0.13 ± 0.02	0.12 ± 0.03	0.401
Docosapentaenoic acid (22:5 n-3)	0.32 ± 0.02	0.28 ± 0.01	0.28 ± 0.02	0.102
Tricosylic acid (23:0)	0.33 ± 0.02	0.26 ± 0.01*	0.24 ± 0.01*	**<0.001**
Lignoceric acid (24:0)	0.64 ± 0.04	0.58 ± 0.03	0.57 ± 0.02	0.297
Nervonic acid (24:1 n-9)	0.98 ± 0.06	0.94 ± 0.06	0.91 ± 0.06	0.779
SFA	34.02 ± 0.50	33.07 ± 0.43	33.74 ± 0.33	0.264
MUFA	24.65 ± 0.69	25.76 ± 0.55	25.50 ± 0.65	0.451
PUFA	41.33 ± 0.77	41.17 ± 0.73	40.76 ± 0.77	0.863
PUFA n-6	36.24 ± 0.79	36.74 ± 0.72	36.73 ± 0.78	0.880
PUFA n-3	5.09 ± 0.3	4.43 ± 0.23	4.03 ± 0.23*	**0.018**
n-6/n-3	7.98 ± 0.46	9.35 ± 0.50	10.23 ± 0.56*	**0.013**
LC-PUFA n-6	9.20 ± 0.31	9.39 ± 0.29	9.74 ± 0.34	0.481
LC-PUFA n-3	4.92 ± 0.30	4.24 ± 0.23	3.87 ± 0.23*	**0.017**
Total Fatty acid concentration (mg/dL)	384.90 ± 17.11	494.31 ± 28.83*	488.50 ± 34.15*	**0.016**

AA arachidonic acid; LC-PUFA, long-chain polyunsaturated fatty acids; MUFA, monounsaturated fatty acids; PUFA, polyunsaturated fatty acids; SFA, saturated fatty acids. Mean ± SEM. Significant differences are indicated by footnote symbols: * Result indicates a significant difference compared to the Control group (*p* < 0.05) (bold face); † Result would indicate significant differences compared to the GDS4 group (*p* < 0.05).

**Table 3 ijms-21-00070-t003:** Fatty acid profile (%) in postmortem brain tissue of Alzheimer’s disease patients and control subjects.

Fatty Acid (%) Respect to Total Fatty Acid Content	Control (*n* = 11)	GDS6 (*n* = 11)	*p*
Lauric acid (12:0)	0.08 ± 0.02	0.05 ± 0.01	0.159
Palmitic acid (16:0)	20.00 ± 0.55	20.19 ± 1.00	0.864
Hexadecanoic acid (16:1 n-9)	0.52 ± 0.03	0.59 ± 0.06	0.342
Palmitoleic acid (16:1 n-7)	0.65 ± 0.04	0.56 ± 0.04	0.141
Margaric acid (17:0)	0.30 ± 0.01	0.31 ± 0.01	0.428
Stearic acid (18:0)	20.55 ± 0.29	19.99 ± 0.30	0.201
Elaidic acid (18:1 trans)	0.05 ± 0.01	0.05 ± 0.01	0.911
Oleic acid (18:1 n-9)	17.58 ± 0.68	18.44 ± 1.08	0.496
Cis-vaccenic acid (18:1 n-7)	4.15 ± 0.21	4.37 ± 0.21	0.462
Linoelaidic acid (18:2 all trans-9,12)	0.01 ± 0.01	0.00 ± 0.00	0.374
Linoelaidic acid (18:2 trans-12)	0.03 ± 0.03	0.00 ± 0.00	0.374
Linoleic acid (18:2 n-6)	0.86 ± 0.09	0.76 ± 0.12	0.478
α-linolenic acid (18:3 n-3)	0.34 ± 0.04	0.34 ± 0.06	0.960
Arachidic acid (20:0)	0.27 ± 0.01	0.27 ± 0.02	0.923
Gondoic acid (20:1 n-9)	0.79 ± 0.10	0.89 ± 0.16	0.568
Eicosadienoic acid (20:2 n-6)	0.23 ± 0.02	0.29 ± 0.02	0.075
Dihomo-γ-linolenic acid (20:3 n-6)	1.08 ± 0.07	1.03 ± 0.08	0.623
Eicosatrienoic acid (20:3 n-3)	0.02 ± 0.02	0.00 ± 0.00	0.374
Arachidonic acid (20:4 n-6, AA)	8.83 ± 0.35	8.60 ± 0.48	0.697
Eicosapentaenoic acid (20:5 n-3, EPA)	0.05 ± 0.04	0.00 ± 0.00	0.174
Behenic acid (22:0)	0.48 ± 0.08	0.65 ± 0.08	0.159
Erucic acid (22:1 n-9)	0.05 ± 0.01	0.09 ± 0.01	0.073
Adrenic acid (22:4 n-6)	4.78 ± 0.14	4.32 ± 0.17	0.051
Docosapentaenoic acid (22:5 n-6, DPA)	1.23 ± 0.10	1.89 ± 0.23*	**0.022**
Docosapentaenoic acid (22:5 n-3)	0.41 ± 0.07	0.26 ± 0.03	0.079
Docosahexaenoic acid (22:6 n-3, DHA)	11.83 ± 0.80	10.77 ± 1.20	0.458
Tricosylic acid (23:0)	0.14 ± 0.07	0.10 ± 0.06	0.679
Lignoceric acid (24:0)	1.05 ± 0.16	1.20 ± 0.29	0.653
Nervonic acid (24:1 n-9)	3.63 ± 0.53	4.00 ± 0.98	0.734
SFA	42.88 ± 0.47	42.77 ± 0.79	0.903
MUFA	27.37 ± 1.32	28.93 ± 2.12	0.525
PUFA	29.67 ± 0.94	28.26 ± 1.41	0.402
TRANS	0.09 ± 0.04	0.05 ± 0.01	0.349
PUFA n-6	17.02 ± 0.37	16.89 ± 0.51	0.832
PUFA n-3	12.65 ± 0.78	11.37 ± 1.14	0.355
n-6/n-3	1.44 ± 0.15	1.69 ± 0.27	0.410
LC-PUFA n-6	16.16 ± 0.33	16.13 ± 0.53	0.966
LC-PUFA n-3	12.31 ± 0.82	11.03 ± 1.19	0.374
Total Fatty Acid concentration (mg/g)	23.82 ± 1.09	22.37 ± 1.62	0.454

AA, arachidonic acid; DHA, docosahexaenoic acid; DPA, docosapentaenoic acid; EPA, eicosapentaenoic acid; LC-PUFA, long-chain polyunsaturated fatty acids; MUFA, monounsaturated fatty acids; PUFA, polyunsaturated fatty acids; SFA, saturated fatty acids; TRANS, trans fatty acids. Mean ± SEM. Significant differences are indicated by footnote symbols: * Result indicates a significant difference compared to the control group (*p* < 0.05) (bold face).

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
