# Peer review of "Decreased Blood Level of MFSD2a as a Potential Biomarker of Alzheimer’s Disease"

_ijms, 2019, doi:10.3390/ijms21010070_

Round 1

Reviewer 1 Report

The authors wanted to examine if DHA transporter Msfd2a can be used as an early marker of AD in humans based on the fact that decreased level of DHA was found in AD brains post mortem. Therefore they analyzed content of fatty acids and Mfsd2a expression in the blood of control and two different AD groups of patients. They found a significant reduction of Mfsd2a and   DHA/omega-3 levels in the more severe AD group as compared to control. However they did not find correlation between mfsd2a and DHA or EPA percentage in the plasma. Further, they analyzed post mortem AD brains and found no significant changes in the level of mfsd2a expression or DHA/ EPA levels.  Disagreement in blood and brain mfsd2a level the authors explained as a consequence of different regulation in the brain.

The hypothesis presented here might be of interest to scientist and clinicians. However, while the title appears to be sound, the study needs to be improved to bring reliable data, good presented and analyzed, that would support the conclusion. Specifically, the number of samples, the quality of representative western blots and unclear statistics are questionable. The language needs some improvement.

There are some examples of issues that need to be addressed:

-In the Abstract ‘post mortem” brains should be used when mentioning that sort of tissue for analysis.

-Throughout the text there are some phrases that need to be replaced by more common scientific expressions. For example, “localization of Mfsd2a in the blood”-meaning (type of) blood cells?, “reduces amyloid trouble”, “intellectual injury”, “non-invasive samples” , “other minor saturated FA” etc…

- Western blot was done from 15 ml of the whole blood. Regarding high abundance of blood proteins (globulins etc..) that can induce non-specific binding and interfere with specific ECL signals (due to similar molecular weight) did authors perform some additional purification?

In the Results section:

-Result 1 use phrase “clinical characteristics” in title and   “demographic characteristics” in the Table 1. It would be better to be more precise and uniformed.  

-As it is much easier to find individuals for control group than AD patients it is strange that some easy-to-obtain parameters from table 1 are missing in Controls and not available for comparison with AD groups. Please explain.

-It is not clear (Table 1) what does ‘a’ mean in statistical analysis. Sine “b” should be significance of GDSs vs control group, and control group has “a” it is a question what Control group was compared to? Or maybe letters mean levels of significance, “a ‘ for p<0.05, “b” for 0.01 ..?

-“As expected DSG6 indicated cognitive severe impairment..” –As we understood it is not something that was expected, or examined but only consequence of criteria for patients sorting. Other parameters from Table 1 are even not mentioned in Results though they can have an impact on hypothesis. For example there is double difference in smoking parameter between two AD groups and no data for Control. The authors should discuss that or mark as a weakness of the study.

Result 2.2 –Western blots are too exposed and unconvincing.  Again, what does “a’ mean on the control? The Result 2.2 is titled “Reduced level of Mfsd2a..” and there is only one sentence about it, while it also contains result of fatty acid analysis and the most of  the text is committed to that. Maybe the title of the section should be modified or these two results separated? FAs in the table should be written with full name for easier following. First part o9f the Table 2 has units in gr/100 gr, second part-mg/dl? Statistics is not clear…

-Figure 2-authors say that there is statistically significant difference between DGS and control in DHA content but they did not mark it on the graph, while again statistics for EPA content on the same graph is not clear…

Fig 3b. In Figure legends, please make correction that samples for western blot are not from blood but form postmortem brain tissue

Table 3 is, similar to Table 2, not uniformed, as units are gr/100gr in the upper part and   mg/g in a lower part. The names of fatty acids are written in a random manner.

Author Response

We really thank reviewer 1 for the insightful and constructive comments which we found very helpful to improve our article.

There are some examples of issues that need to be addressed:

Point 1: In the Abstract ‘post mortem” brains should be used when mentioning that sort of tissue for analysis.

According your suggestion it was included in the abstract, page 1, line 21.

Point 2: Throughout the text there are some phrases that need to be replaced by more common scientific expressions. For example, “localization of Mfsd2a in the blood”-meaning (type of) blood cells?, “reduces amyloid trouble”, “intellectual injury”, “non-invasive samples” , “other minor saturated FA” etc…

“Localization of Mfsd2a in the blood”  was replaced by “MFSD2a in the whole blood”(Abstract section, pag 1, line 19 and line 29, legend of figure 1, pag 3 line 117, discussion section page 9, line 177).

“Reduces amyloid trouble” was replaced by “amyloid burden” in Introduction section, page 2, line 53.

“Intellectual injury” was replaced by “intellectual disability” in Introduction section, page 2, line 70.

“Non-invasive samples” was replaced by “samples obtained non-invasively” page 2, Line 83.

“Other minor saturated FA”. We specified the minor fatty acids. We included “other minor saturated FA (22:0 and 23:0) decreased” in Results section page 4 line 128

- Western blot was done from 15 ml of the whole blood. Regarding high abundance of blood proteins (globulins etc.) that can induce non-specific binding and interfere with specific ECL signals (due to similar molecular weight) did authors perform some additional purification?

Western Blot analysis was not done from 15 mL of the whole blood, but from 15 µL of the whole blood diluted 20 times in cell lysis buffer, as it was indicated in the paper. The whole blood was diluted in cell lysis buffer, centrifuged and supernatant collected. Only 11µg of protein from the blood supernanat was used for the western-blot analysis (Material and Method section, page 9, paragraph 289).

In the Results section:

-Result 1 use phrase “clinical characteristics” in title and   “demographic characteristics” in the Table 1. It would be better to be more precise and uniformed. 

Following your suggestion, we used the same words “demographic characteristics” in both title of section 1 of results (page 2, line 88) and headline of Table 1.

-As it is much easier to find individuals for control group than AD patients it is strange that some easy-to-obtain parameters from table 1 are missing in Controls and not available for comparison with AD groups. Please explain.

All samples (blood and postmorten brain tissue) were provided from the Brain Biobank of the Hospital of Murcia Region (Spain). It is very difficult to collect such brain samples, and we decided to use available resources from biobanks. These clinical data were the only one available. We agree that it would be interesting to have access to that information in the controls too, but this was available for us. This limitation was included in page 3, line 94.

-It is not clear (Table 1) what does ‘a’ mean in statistical analysis. Sine “b” should be significance of GDSs vs control group, and control group has “a” it is a question what Control group was compared to? Or maybe letters mean levels of significance, “a ‘ for p<0.05, “b” for 0.01 ..?

 In order to avoid confusion we have indicated in the legend of Tables and figures “Different superscript letters indicate significant differences between groups (P <0.05).”. In the statistical analyses the three groups are compared at the same time among them; in table 1 we indicated with “b” superscript that GDS-4 and GDS-6 were similar among them, but both different respect to control “a”. 

-“As expected DSG6 indicated cognitive severe impairment..” –As we understood it is not something that was expected, or examined but only consequence of criteria for patients sorting.

We agree and we have eliminated “as expected” on page 2 line 91

Other parameters from Table 1 are even not mentioned in Results though they can have an impact on hypothesis. For example there is double difference in smoking parameter between two AD groups and no data for Control. The authors should discuss that or mark as a weakness of the study.

We agree and we have included in results section page 2 lines 92 to 95 “No differences were found between the groups of AD patient in diabetes and hypertension. However, there is a trend in smoking differences between Alzheimer's patients although this information from control group was not available due to a limitation on the Biobank data provided from such samples.”

Result 2.2 –Western blots are too exposed and unconvincing. 

The image is an example of western blot and is not manipulated in terms of color, brightness and anything else. The overexposing is detected by the software in pink colour, and as you can see in the image, there is not pink colour in the image neither in the rest of samples analyzed.

Again, what does “a’ mean on the control?

As we have explained before, in order to avoid confusion we wrote: “Different superscript letters indicate significant differences between groups (P <0.05).”

The Result 2.2 is titled “Reduced level of Mfsd2a..” and there is only one sentence about it, while it also contains result of fatty acid analysis and the most of  the text is committed to that. Maybe the title of the section should be modified or these two results separated?

Following your suggestion, we have modified the title of section result 2.2: “Reduced level of blood MFSD2a in Alzheimer’s disease patients and impact on fatty acid profile in serum (pag 3, line 111)

FAs in the table should be written with full name for easier following.

According to your suggestion, full names of fatty acids were included on both table 2 and table 3.

First part o9f the Table 2 has units in gr/100 gr, second part-mg/dl? Statistics is not clear…

Fatty acids are usually expressed as g/100g of fat. Table 2 shows fatty acid results expressed as g/100g of fatty acids except for the line of “Total Fatty acids” that indicate the  total absolute amount of fat (mg/dL). In order to avoid confusion, “Total Fatty acids (mg/dL)”  data was written at the end of the tables and not in the medium of them.

-Figure 2-authors say that there is statistically significant difference between DGS and control in DHA content but they did not mark it on the graph, while again statistics for EPA content on the same graph is not clear…

We indicated in page 5 line 139 that “In fact, there was a clear trend towards a decrease of the DHA percentage in serum of GDS4 and GDS6 groups (P=0.062).”. The ANOVA test was statistically significant for LC-PUFA n-3 and EPA but showing a clear trend of decrease for DHA, because p value was lower than 0.1 that it is considered the cut-off point of significant trend.

Since p value of DHA in the ANOVA was higher than 0.05 (p=0.062), superscript letters cannot be used in the figure 2 for DHA. We also mentioned, in page 5, line 140 that we also made a t-test between both (GDS-4 + GDS-6 vs. controls) and the results were significantly reduced in Alzheimer patients for DHA in serum.

Fig 3b. In Figure legends, please make correction that samples for western blot are not from blood but form postmortem brain tissue.

We have included “postmortem brain tissue” in the legend of Figure 3b.

Table 3 is, similar to Table 2, not uniformed, as units are gr/100gr in the upper part and   mg/g in a lower part. The names of fatty acids are written in a random manner.

Similar to Table 2, we have moved “Total Fatty acids (mg/dL)” data to the end of the Table 3.

Reviewer 2 Report

Control subjects are significantly younger than AD patients, which is a major issue in the present illness.

MFSD2A is localized in endothelial cells in the brain. Is there possibility that AD induced endothelial injury, which can be a cause of decline of MFSD2A levels in the serum?

In the Introduction and Discussion, authors discuss the biological function, including resolvin D, protectins, and maresins, of DHA. However, it is necessary to clearly state whether the effects were

In table 2, levels of total fatty acids were 100mg/dL higher in AD patients than that in normal subjects. Which sorts of fatty acids were higher in AD?

In figure 1 and 2, what is ‘a’ and ‘b’?

In Figure 3 What is ‘chart area’?

Author Response

We really thank reviewer 2 for the insightful and constructive comments which we found very helpful to improve our article.

Control subjects are significantly younger than AD patients, which is a major issue in the present illness.

As you realize, it is very difficult to get access to that samples, and then we had to work with the samples collected from the biobank. Nevertheless, all subjects were above a mean of 65 years as indicated in the Table 1. Patients from GDS-4 and GDS-6 were similar in age but these groups followed a gradient on the MFSD2a and DHA results in blood respect to controls. Thus, the pathology seems to have the highest effect on the parameters analyzed. We added in the Discussion section page 4, lines 214 that: “Despite the differences on age between groups, patients from GDS-4 and GDS-6 had similar age but a gradient on lower MFSD2a and DHA in blood respect to controls. Thus, the pathology seems to have the highest effect on the parameters analyzed.”

MFSD2A is localized in endothelial cells in the brain. Is there possibility that AD induced endothelial injury, which can be a cause of decline of MFSD2A levels in the serum?

We found too speculative to indicate that the reason for the decline of MFSD2a levels in the serum could be due to an endothelial injury. Indeed we did not found differences among groups in the level of MFSD2a in brain.

In the Introduction and Discussion, authors discuss the biological function, including resolvin D, protectins, and maresins, of DHA. However, it is necessary to clearly state whether the effects were

According your suggestion, we added in the discussion page 9, line 248 : “E-series resolvins (RvEs) derived from eicosapentaenoic acid (EPA), and D-series resolvins (RvDs), protectin/neuroprotectin (PD/NPD) and maresins (MaRs) derived from DHA have potent local pro-resolving actions” (Xu, Tan et al. 2013; Buckley, Gilroy et al. 2014; Zhu, Wang et al. 2016). However, we did not find significant differences in the percentages of DHA or EPA in the hippocampus of the postmortem subjects.

Buckley, C. D., D. W. Gilroy, et al. (2014). "Proresolving lipid mediators and mechanisms in the resolution of acute inflammation." Immunity 40(3): 315-327.

Xu, M. X., B. C. Tan, et al. (2013). "Resolvin D1, an endogenous lipid mediator for inactivation of inflammation-related signaling pathways in microglial cells, prevents lipopolysaccharide-induced inflammatory responses." CNS Neurosci Ther 19(4): 235-243.

Zhu, M., X. Wang, et al. (2016). "Pro-Resolving Lipid Mediators Improve Neuronal Survival and Increase Abeta42 Phagocytosis." Mol Neurobiol 53(4): 2733-2749.

In table 2, levels of total fatty acids were 100mg/dL higher in AD patients than that in normal subjects. Which sorts of fatty acids were higher in AD?

There were higher amount in concentration values of practically all the fatty acids, and for this reason the total amount of FA is higher in the AD groups. As it was shown in the table 2 using in percentage values, all fatty acid indexes (saturated, monounsaturated and polyunsaturated) were similar in the three experimental groups, thus a similar increase in absolute concentration values of these fatty acids indices was observed in AD patients except for the n-3 PUFA.

In figure 1 and 2, what is ‘a’ and ‘b’?

In order to avoid confusion we have indicated in the legend of Tables and figures “Different superscript letters indicate significant differences between groups (P <0.05)”. In the statistical analyses the three groups are compared at the same time among them; in table 1 we indicated with “b” superscript that GDS-4 and GDS-6 were similar among them, but both different respect to control “a”. 

In Figure 3 What is ‘chart area’?

“Chart area” is an error in the figure due to the template. It was removed.

Round 2

Reviewer 1 Report

The authors made weak effort to improve the paper. Even when the suggestions of the reviewers were adopted, very little was done to make the changes meaningful in the context of the text. Some questions are not adequately responded.  The explanation of the statistical results shown in the figures and tables is particularly worrying.

Author Response

Specific answers to Reviewer 1.

The authors made weak effort to improve the paper. Even when the suggestions of the reviewers were adopted, very little was done to make the changes meaningful in the context of the text. Some questions are not adequately responded. The explanation of the statistical results shown in the figures and tables is particularly worrying.

 We really thank reviewer 1 for his/her insightful and constructive comments, which we found very helpful to improve our article. We have re-edited carefully the manuscript and it has been corrected by an English native to avoid grammatical and orthographical mistakes. All the changes were clearly highlighted using the "Track Changes" function in Microsoft Word, as indicated by the editor.

We really apologize for the misunderstanding with the statistical symbols used in the previous version. This is an important issue. To avoid any confusion we have used in the revised version another type of statistical symbols. We have deleted the superscript letters in all tables and figures and they were substituted by other type of footnote symbols. We indicated in the legend of all Tables and Figures that: “Significant differences are indicated by footnote symbols: *Result indicates significant difference compared to Control group (P < 0.05); †Result indicates significant differences compared to GDS4 group (P < 0.05).” We hope that now the results can be easier to be followed by the readers.

We also re-wrote the headlines of Table 2 and Table 3 to better indicate the fatty acid units. The Fatty acid profile of samples is commonly expressed as fatty acid percentages (%) of each fatty acid respect to the total fatty acids. The reason to work on percentage values with fatty acids is due to the fact that fatty acids are extracted with apolar solvents that evaporate easily and then this may affect to the absolute amount quantified by gas chromatography. For this reason practically all the papers work on fatty acid percentage values. However, we have also indicated in the last row, of both tables, the total absolute concentration of fatty acids in the tissue. In case one researcher would like to know the absolute concentration of each fatty acid, only would have to apply the percentage of the specific fatty acid to the total amount of fatty acids reported in the last row of the tables. We added in the page 11, lines 347-350 from the Material and Methods Section, Section 4.4 Fatty acid analysis, that:

 “We provide the fatty acid profile as percentage values of each fatty acid respect to the total fatty acids. In addition, the absolute concentration of total fatty acids in the samples is specified at the end of the tables. Multiplying the percentage of the specific fatty acid by the total amount of fatty acids can be estimated the absolute concentration of each fatty acid analyzed.”

The positions of fatty acid names in Tables 2 and 3 have been modified according to the carbon length chain.

We really appreciate all the comments and it would be a pleasure to answer any doubt from the reviewer.

Reviewer 2 Report

Manuscript was improved, and I have no further comments.

Author Response

We really thank reviewer 2 for his/her insightful and constructive comments, which we found very helpful to improve our article.

Round 3

Reviewer 1 Report

The authors improved the manuscript as required.